# Magnetic Solid-Phase Extraction of Dichlorodiphenyltrichloroethane and Its Metabolites from Environmental Water Samples Using Ionic Liquid Modified Magnetic Multiwalled Carbon Nanotube/Zeolitic Imidazolate Framework-8 as Sorbent

**DOI:** 10.3390/molecules24152758

**Published:** 2019-07-29

**Authors:** Xiaodong Huang, Yanan Liu, Huifang Liu, Guangyang Liu, Xiaomin Xu, Lingyun Li, Jun Lv, Zhongxiao Liu, Wenfeng Zhou, Donghui Xu

**Affiliations:** 1Institute of Vegetables and Flowers, Chinese Academy of Agricultural Sciences, Key Laboratory of Vegetables Quality and Safety Control, Laboratory of Quality & Safety Risk Assessment for vegetable Products, Ministry of Agriculture and Rural Affairs of China, Beijing 100081, China; 2Department of Applied Chemistry, China Agricultural University, Beijing 100193, China; 3School of Life Science and Food Engineering, Hebei Engineering University, Handan 056000, China

**Keywords:** dichlorodiphenyltrichloroethane, magnetic solid phase extraction, zeolitic imidazolate framework, ionic liquid, environmental samples

## Abstract

As persistent organic pollutants, dichlorodiphenyltrichloroethanes (DDTs) and their metabolites pose considerable risks to human health and the environment. Therefore, monitoring DDTs in the environment is essential. Here, we developed a green, simple, and effective magnetic solid-phase extraction (MSPE) method coupled with gas chromatography tandem triple-quadrupole mass spectrometry to determine the DDT content of environmental water samples. A magnetic ionic liquid (IL) adsorbent was developed based on a modified magnetic multiwalled carbon nanotube/zeolitic imidazolate framework-8 (MM/ZIF-8/IL), synthesized by immobilizing the IL on the surface of MM/ZIF-8. We confirmed successful synthesis of MM/ZIF-8/IL by material characterization, and our results suggested that the MM/ZIF-8/IL had a high Brunauer–Emmett–Teller surface area (159.9 m^2^ g^−1^), good thermostability (<800 °C), and a high degree of superparamagnetism (52.9 emu g^−1^). Several experimental conditions affecting the MSPE efficiency were optimized. Under the best conditions, good detection linearity was achieved (0.5–500 µg L^−1^) with determination coefficients ranging from 0.9927 to 0.9971. The lower limits of detection (0.0016–0.0072 µg L^−1^) also had good precision, having an intraday relative standard deviation (RSD) ≤ 6.5% and an interday RSD ≤ 8.9%. Finally, we used the as-developed method to determine DDT levels in environmental water samples.

## 1. Introduction

Dichlorodiphenyltrichloroethanes (DDTs) are organochlorine pesticides that include *o,p′*-DDT; *p,p′*-DDT; and their metabolites *o,p′*-DDD, *p,p′*-DDD, *o,p′*-DDE, and *p,p′*-DDE, which have been used extensively in pest control programs over the last century. Owing to their carcinogenicity, teratogenicity, bioaccumulation, and degradation resistance, DDTs pose considerable risks to human health and the environment [1]. Thus, use of DDTs has been banned, in accordance with the Stockholm Convention, for many years. However, it has been reported that DDTs are present at a concentration of 15 ng L^−1^ in the aqueous environment around Beijing, China [2]. Thus, it is essential to develop a simple, sensitive, and reliable analytical technique to enable continuous monitoring of trace or ultratrace levels of DDTs in environmental water samples.

Sample pretreatment is an important step in the analysis of pesticide residues, particularly for determining persistent organic pollutants at the trace or ultratrace level. To date, various sample pretreatment techniques have been developed to analyze DDTs, such as solid-phase extraction (SPE), solid-phase microextraction, quick easy cheap effective rugged safe (QuEChERS) techniques, and magnetic solid-phase extraction (MSPE) [3,4,5,6]. As a relatively new type of SPE, MSPE is based on the use of magnetic sorbents, which can be easily retrieved under a magnetic field. This method avoids tedious or time-consuming filtration and centrifugation operations, such that MSPE has been identified as a simple and rapid sample pretreatment technique [7]. Recently, several magnetic adsorbents have been prepared for applications as MSPE materials to analyze DDTs, such as magnetic carbon nanomaterials and magnetic metal–organic frameworks (MOFs) [8,9].

Multiwalled carbon nanotubes (MWCNTs) are composed of many layers of seamless rolled graphite sheets. Owing to their remarkable mechanical, thermal, electronic, and chemical properties, MWCNTs have drawn considerable attention for sample pretreatments [10]. Recently, MWCNTs have reportedly been used to prepare magnetic adsorbents for MSPE of antibiotics, estrogen, mycotoxins, metal ions, environmental pollutants, and pesticides [11,12,13,14,15,16]. As reported, magnetic MWCNTs (Fe_3_O_4_–MWCNTs, M-MWCNTs) combined with MOFs can be used as adsorbents for organochlorine pesticides and organophosphorus pesticides from aqueous samples [17,18]. Therefore, magnetic MWCNTs integrated with MOFs show good potential for enriching persistent organic pollutants in environment samples.

Zeolitic imidazolate frameworks (ZIFs), produced by coordination between imidazolate organic linkers and metal ions (Zn^2+^ and Co^2+^), are regarded as a new subclass of MOFs [19]. On the basis of their excellent physical and chemical properties, thermal stability, and adsorption capacity, ZIFs are microporous with uniformly structured cavities and large surface areas [20]. ZIF-8 is formed by coordination between Zn^2+^ and 2-methylimidazole and has attracted considerable attention owing to its hierarchical porous structure, high surface area, easy preparation, and cost effectiveness. This material system has been widely used in gas storage, drug delivery, sensing, catalysis, and matrix membranes [21,22,23,24,25]. Furthermore, ZIF composites synthesized by integration with other functional materials, including carbon nanomaterials, magnetic nanospheres, and organic–inorganic materials, promote the adsorptive properties of different target analytes [26,27].

Ionic liquids (ILs) are a class of liquid molten salts with melting points at room temperature, which are usually formed by association of inorganic anions and large organic cations. Compared with traditional organic solvents, ILs are regarded as “green” solvents owing to their unique chemical and thermal stabilities, extremely low vapor pressures, nonflammability, and high selectivity towards analytes [28]. Hence, ILs have drawn considerable attention in catalysis, adsorption and separation, and modification [29,30,31]. ILs can be used as a modifier on a support material to introduce different interactions between an adsorbent and a target analyte, which include ion-exchange, hydrophobic, electrostatic, hydrogen bonding, and π–π interactions [32]. To date, some IL-modified ZIFs have been prepared for sample pretreatment [33]. However, no ILs have been reported as modifying agents of magnetic MWCNT/ZIF-8 or used as an adsorbent of MSPE for DDTs.

The purpose of this work is to develop a simple, sensitive, and reliable analytical method based on an IL-modified magnetic MWCNT/ZIF-8 composite as an adsorbent for MSPE of DDTs from environmental water samples. In a previous study, we used magnetic MWCNT/ZIF-8 for fast and effective adsorption and removal of organophosphorus pesticides from environmental samples [18]. However, the adsorption affinity of the magnetic MWCNT/ZIF-8 for DDTs was not measured and the as-developed method was not appropriate for monitoring trace or ultratrace levels of DDTs. After preparation of Fe_3_O_4_/MWCNT/ZIF-8/IL (MM/ZIF-8/IL), we conducted material characterization experiments to confirm its morphology, structure, and properties. Furthermore, we evaluated the factors affecting the MSPE efficiency by a single-factor optimization design. Finally, we used the optimized MM/ZIF-8/IL-based MSPE technique to determine DDT levels in real environmental water samples.

## 2. Results and Discussion

### 2.1. Characterization of MM/ZIF-8/IL

The microtopography of MM/ZIF-8/IL was characterized by SEM. As shown in Figure 1A, the composite had a highly porous block-shaped structure with masses of aggregated MWCNTs, microspheres, or nanoparticles. The unique structure of the MM/ZIF-8/IL should facilitate adsorption.

The successful preparation of MM/ZIF-8/IL was verified by FT-IR and XRD. The FT-IR spectra of MM/ZIF-8/IL and other synthetic magnetic materials are shown in Figure 1B. The spectrum of MM/ZIF-8/IL showed an adsorption band at 521 cm^−1^, which corresponded to the Fe–O–Fe vibration from magnetite, and a band at 1337 cm^−1^ was attributed to the in-plane bending of CH_2_ in the MWCNTs [34]. The bands between 992 and 1309 cm^−1^ and at 1420 cm^−1^ corresponded to the Zn–N vibration of ZIF-8 [18]. The C–N vibrations of the imidazole ring were located at 2928 cm^−1^, and the C–H vibrations of saturated hydrocarbon were at 3133 and 3339 cm^−1^ [35]. The XRD patterns of the obtained materials are shown in Figure 1C. Characteristic peaks at 21.2°, 35.2°, 41.6°, 50.8°, 63.3°, 67.5°, and 74.4° were well matched with those of the Fe_3_O_4_ crystal structure, and the peaks at 12.0°, 14.8°, 17.1°, and 19.0° corresponded to the crystal structure of ZIF-8 [36]. Comparison of the XRD patterns of synthetic magnetic materials showed no clear differences between their diffraction peaks. These characterization results for the synthetic materials indicate successful preparation of the MM/ZIF-8/IL composite.

We used a vibrating sample magnetometer (VSM), N_2_ adsorption, and TGA to confirm the magnetic properties, porosity, and thermostability of the obtained materials, respectively. As illustrated in Figure 1D, the saturation magnetization values of M-MWCNTs, MM/ZIF-8, and MM/ZIF-8/IL were 66.9, 56.0, and 52.9 emu g^−1^, respectively. The remanence and coercivity values of the three materials were zero, which indicated superparamagnetic properties enabling easy collection under a magnetic field. The N_2_ adsorption–desorption isotherm of MM/ZIF-8/IL is shown in Figure 1E. The Brunauer–Emmett–Teller surface area and the total pore volume were 159.9 m^2^ g^−1^ and 0.49 cm^3^ g^−1^, respectively. These results suggest an acceptance surface area and pore volume of MM/ZIF-8/IL, which should enable adsorption of DDTs. We performed TGA to study the thermostability of the MM/ZIF-8/IL from room temperature to 800 °C. As shown in Figure 1F, 6.9% and 14.1% weight losses occurred at 500 and 800 °C, respectively. The probable cause of this weight loss was evaporation of water contained in the nanochannel of the ZIF-8. The TGA results indicated that MM/ZIF-8/IL was thermally stable.

### 2.2. Optimization of the Extraction Conditions

To optimize the extraction conditions, we performed single-factor experiments to investigate several parameters affecting the DDT extraction performance of the MSPE, including the type and content of IL, the amount of sorbent, extraction time, ionic strength, and pH of the sample solution.

The structure of ILs determines their physicochemical characteristics, which can influence the pesticide residue extraction efficiency [37]. To determine an appropriate IL and confirm its optimal content, we examined the hydrophilic ILs [HMIM]NTF_2_, [HMIM]PF_6_, [OMIM]NTF_2_, and [OMIM]PF_6_ as candidates for modifying MM/ZIF-8. After modification, the four types of MM/ZIF-8/IL were used to adsorb DDTs. As illustrated in Figure 2A, [HMIM]PF_6_ promoted extraction of the DDTs most effectively and was therefore selected as the optimal IL for immobilization. We then tested different mass ratios of MM/ZIF-8 to [HMIM]PF_6_ of 0.2:1, 0.5:1, 1.0:1, 1.5:1, and 2.0:1 to confirm the optimum composition of MM/ZIF-8/IL. As shown in Figure 2B, the recoveries of DDTs were best when the ratio was set to 1.0:1. However, an excess or insufficient amount of MM/ZIF-8 resulted in lower recovery. An excess of the IL likely reduces the dispersibility of MM/ZIF-8/IL and a lower IL content might not be able to extract the DDTs. Therefore, the optimum mass ratio of MM/ZIF-8 to [HMIM]PF_6_ was set to 1.0:1.

Sorbents play an important role in extraction processes. To investigate the influence of the amount of sorbent on the extraction efficiencies for target analytes, a series of MM/ZIF-8/IL masses ranging from 2 to 10 mg were added to 10 mL sample solutions to evaluate the extraction performance. The recoveries of all analytes increased as the sorbent dose was increased from 2 to 4 mg and then remained nearly constant with further increases in the mass of the adsorbent (Figure 2C). Herein, the optimum dosage of MM/ZIF-8/IL was set to be 4 mg.

The extraction time affects the partition of the analytes between the adsorbent and sample solution. To clarify the effects of the extraction time on MSPEs for DDTs, we investigated five different vortex mixing times (0.5–2.5 min) for the extraction. The recoveries of DDTs increased as the adsorption time was increased from 1.0 to 1.5 min and remained relatively constant with further increases in the extraction time (Figure 2D). Hence, the optimum extraction time was set at 1.5 min.

The salt effect is another parameter affecting extraction efficiency. Owing to the salt-out effect, the solubility of target compounds can be altered in the matrix solution [38]. To confirm the effects of salt on the MSPE, we performed several experiments with the addition of five different amounts of NaCl (0%, 1%, 2%, 5%, and 7%, w/v) to the sample. As shown in Figure 2E, the recoveries increased as the NaCl content was increased from 0% to 5% but decreased with further increases in NaCl content. Therefore, 2% (w/v) NaCl was used as the ideal salt content for MSPE.

The sample solution pH also influences the extraction efficiency by changing the surface charge of the sorbent and/or chemical form of the analyte. To investigate the effects of the solution pH on the MSPE performance, the sample solution pH was adjusted from 4.0 to 10.0 with a HCl and NaOH solution. As illustrated in Figure 2F, the recoveries increased as the pH value was increased from 4.0 to 7.0. However, the recoveries decreased with further increases of the pH. The probable cause for this decrease was that the DDTs became unstable under alkaline conditions [39]. As a result, a solution pH of 7.0 was used for subsequent experiments.

### 2.3. Optimization of the Desorption Conditions

A desorption solvent plays an important role in promoting the complete desorption of the target analytes from the sorbent. To study the effects of desorption solvents on MSPE performance, we selected acetone, methanol, ethyl acetate, acetonitrile, and *n*-hexane as potential desorption solvents. As shown in Figure 3A, ethyl acetate gave the best recoveries. The effect of eluent volume was also studied with volumes over the range of 0.4–1.5 mL, and these results indicated that 0.8 mL was the optimal eluent volume (Figure 3B). The effects of desorption time on the MSPE efficiency were also investigated by varying the vortex time from 0.5 to 2.5 min. As shown in Figure 3C, a desorption time of 2.0 min was satisfactory for eluting DDTs from the sorbent. Thus, the optimized process for eluting DDTs involved the use of 0.8 mL of ethyl acetate with vortexing for 2.0 min, and the desorption operation was repeated once.

### 2.4. Method Validation

The proposed method was validated for linearity, limit of detection (LOD), and repeatability under the optimized conditions. The linearity was determined by analyzing ultrapure water spiked with standard solutions of the DDTs at 0.5–500 µg L^−1^ and plotting the peak areas versus the concentrations. The validation results are shown in Table 1. Good linearities for DDTs were achieved over the concentration range of 0.5–500 µg L^−1^, and the determination coefficients (*R*^2^) ranged from 0.9927 to 0.9971. The LODs were 0.0016–0.0072 µg L^−1^, calculated at a signal/noise ratio of 3. The repeatability, defined as the method precision, was confirmed by measuring the relative standard deviations (RSDs) in the samples spiked with the standard solution at 10 µg L^−1^ with six replicates. The RSD values for intraday and interday precision were in the range of 1.0–6.5% and 1.0–8.9%, respectively. Comprehensive analyses of the above data indicate that the proposed method has high sensitivity, a wide linear range, and good repeatability.

### 2.5. Comparison of the MM/ZIF-8/IL-Based Method with other Reported Methods

The performance of the MM/ZIF-8/IL-based method was compared with other methods, and the comparison data are listed in Table 2. By comprehensive analyses of the sorbent amount, extraction time, type and volume of eluent, desorption time, analytical range, and LOD, the as-developed method showed better analytical performance and operational efficiency than the other techniques. This method is also simple and environmentally friendly, considering the small volume of eluent and the MSPE-based process.

### 2.6. Real Sample Analysis

The practical performance of the proposed method was investigated with three real water samples. The samples were analyzed directly and then spiked with two concentrations of DDTs (10 and 100 μg L^−1^) before MSPE. As listed in Table 3, the recoveries of DDTs from the three samples were in the range of 72.6–97.5% with RSDs lower than 4.1%, which indicates good utility for determining DDT levels in environmental water samples. No DDTs were found in the unspiked real water samples. The extracted total ion chromatograms of DDTs obtained from the tap water sample are illustrated in Figure 4.

## 3. Materials and Methods 

### 3.1. Reagents and Materials

The standards of *o,p′*-DDD, *p,p′*-DDD, *o,p′*-DDE, *p,p′*-DDE, *o,p′*-DDT, and *p,p′*-DDT were in liquid form and were in concentrations of 1000 mg L^−1^. All of these standards were supplied by the Agro-Environmental Protection Institute, Ministry of Agriculture (Tianjin, China). Standard mixture stock solution (100 mg L^−1^) was prepared in methanol and stored at 4 °C in darkness. HPLC-grade organic solvents, including methanol, acetone, acetonitrile, ethyl acetate, and *n*-hexane, were purchased from Sigma-Aldrich (St. Louis, Missouri, USA). The MWCNTs (3–5 nm inner diameter (id), 50 µm long, 95% purity), ferric chloride hexahydrate, ferrous chloride tetrahydrate, zinc sulfate heptahydrate, 2-methylimidazole, and ammonium hydroxide (mass fraction 28%) were obtained from Aladdin Industrial Co. (Shanghai, China). 1-Hexyl-3-methylimidazolium bis(trifluoromethanesulfonyl)imide ([HMIM]NTF_2_), 1-hexyl-3-methylimidazolium hexafluorophosphate ([HMIM]PF_6_), 1-octyl-3-methylimidazolium bis(trifluoromethanesulfonyl)imide ([OMIM]NTF_2_), and 1-octyl-3-methylimidazolium hexafluorophosphate ([OMIM]PF_6_) were purchased from Shanghai Yuanye Bio-Technology Co. (Shanghai, China). Analytical-grade anhydrous ethanol, sodium chloride (NaCl), hydrochloric acid (HCl), and sodium hydroxide (NaOH) were supplied by the Beijing Chemical Reagents Co. (Beijing, China).

### 3.2. Instruments

Gas chromatography tandem mass spectrometry (GC-MS/MS) analyses were conducted using a Shimadzu GC-2010 plus equipped with a Shimadzu TQ8040 triple-quadrupole MS (Shimadzu, Kyoto, Japan). A capillary column (Rtx-5Ms, 30 m long × 0.25 mm id, 0.25 μm film thickness) from RESTEK was used for the separations. The injector temperature was 250 °C, and the detector temperature was 300 °C. High-purity helium (99.999%) was adopted as the carrier gas with a constant flow rate of 1 mL min^−1^. The program of the column temperature was as follows: the temperature was initially maintained at 40 °C for 4 min, then ramped up to 125 °C at a rate of 25 °C min^−1^, and finally increased to 300 °C at a rate of 10 °C min^−1^ and held for 6 min. Splitless mode was selected with an injection volume of 1.0 μL. The specific multiple reaction monitoring (MRM) transitions for DDTs and the other chromatographic parameters are presented in Table 4.

Scanning electron microscopy (SEM) images were obtained using a JSM-6300 electron microscope (JEOL, Tokyo, Japan). Fourier-transform infrared (FT-IR) spectroscopy was performed using an FT-IR-8400 spectrometer (Shimadzu, Kyoto, Japan). X-ray powder diffraction (XRD) measurements were conducted on a D8 Advance diffractometer (Bruker, Karlsruhe, Germany). The magnetic properties were confirmed using a Versalab-7410 VSM (Lake Shore, Columbus, OH, USA). The N_2_ adsorption–desorption isotherms were obtained using an ASAP2460 Surface Area and Porosity Analyzer (Micromeritics, Norcross, Georgia, USA). TGA was conducted using an STA449F3 simultaneous thermal analyzer (NETZSCH, Selbu, Germany).

### 3.3. Sample Preparation

Three real environmental water samples were adopted for evaluation. For the river water, the sample was gathered from the Liangshui River, Beijing, China. For the well water, the collection place was Langfang City, Hebei Province, China. For the tap water, the sample was obtained from a water tap after flowing for 5 min in our laboratory. After collection, all the real samples were filtered through a 0.45 µm micropore membrane and stored at 4 °C in brown glass bottles before usage.

### 3.4. Preparation of MM/ZIF-8/IL

The component materials of MM/ZIF-8/IL, including Fe_3_O_4_/MWCNTs and Fe_3_O_4_/MWCNTs/ZIF-8 (MM/ZIF-8), were prepared by strictly following our former reported method [34].

For the composite of MM/ZIF-8/IL, the synthesis procedure was performed in accordance with a previously reported method with slight modifications [35]. Briefly, different quantities of MM/ZIF-8 (0.2, 0.5, 1.0, 1.5, and 2.0 g) were dispersed into 6 mL of ethanol solution that contained 1.0 g of IL. The mixtures were stirred for 30 min at room temperature. The products were then gathered with the use of a magnet and washed three times with ethanol. Finally, the magnetic composites were dried at 60 °C in a vacuum oven for 24 h.

### 3.5. Extraction Procedure

The MSPE process was as follows: First, 4 mg of MM/ZIF-8/IL was added to a 15 mL centrifuge tube that contained 10 mL of the sample solution (NaCl, 2%, *m*/*v*; pH 7.0), followed by 1.5 min of vortex mixing to facilitate the adsorption. After that, a magnet was placed on the outside bottom of the centrifuge to collect the adsorbent, and the supernatant was then discarded. Subsequently, 0.8 mL of ethyl acetate was added to the tube to elute the target analytes under vortexing for 2.0 min. The sorbent was collected again, and the supernatant eluent was then transferred to a new vial. The same desorption procedure was conducted one more time. Finally, the combined eluent was concentrated by drying under nitrogen at room temperature and then dissolved in 0.5 mL of ethyl acetate. The eluate (1.0 μL) was analyzed by GC-MS/MS. A typical procedure for the proposed MSPE method is shown in Figure 5.

## 4. Conclusions

In this study, an adsorbent was synthesized using ionic-liquid-modified magnetic MWCNTs/ZIF-8. The prepared composite combined the merits of MWCNTs, magnetic MOFs, and IL, and as a result, revealed good adsorption ability and good selectivity. The obtained MM/ZIF-8/IL was used as an adsorbent for MSPE of DDTs from environmental water samples. Combined with GC-MS/MS, the MM/ZIF-8/IL-based method provided a wide linear range, excellent sensitivity, good accuracy, and good precision. In comparison with reported methods for the determination of DDTs, the proposed method is convenient, sensitive, and environmentally friendly. The MM/ZIF-8/IL composite concept could be extended to other IL-modified MOFs and applied for the development of different sample pretreatment techniques.

## Figures and Tables

**Figure 1 molecules-24-02758-f001:**
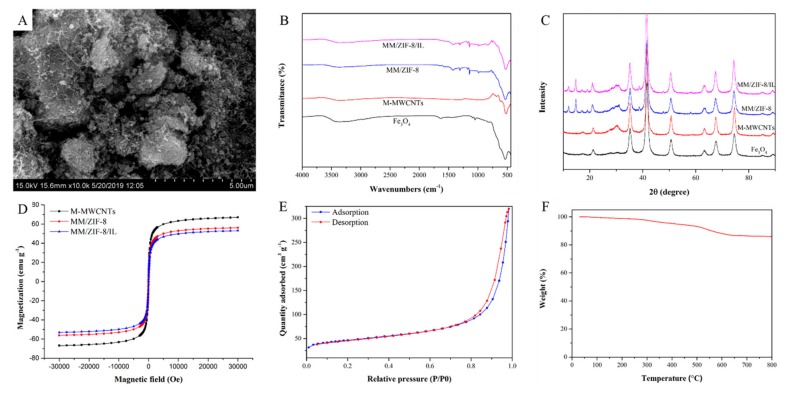
Characterization of the synthetic materials. (**A**) SEM image of magnetic multiwalled carbon nanotube/zeolitic imidazolate framework-8 ionic liquid (MM/ZIF-8/IL); (**B**) FT-IR spectra, (**C**) XRD patterns, and (**D**) magnetic curves of the synthetic materials; (**E**) N_2_ adsorption–desorption isotherm; and (**F**) TGA curve of the MM/ZIF-8/IL.

**Figure 2 molecules-24-02758-f002:**
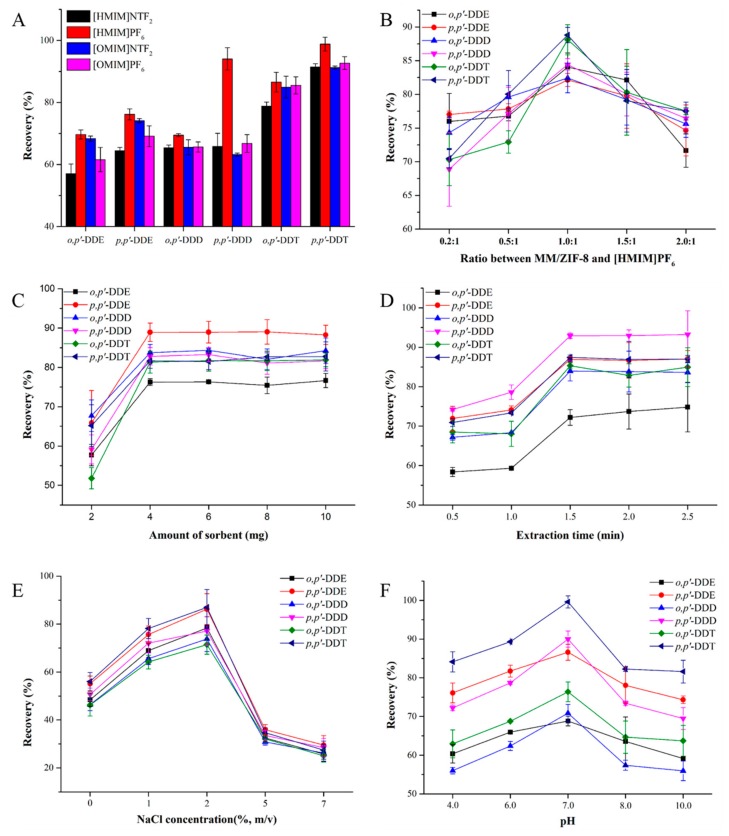
Effects of different parameters on the extraction efficiency of dichlorodiphenyltrichloroethanes (DDTs): (**A**) type of IL, (**B**) mass ratio of MM/ZIF-8 and [HMIM]PF_6_, (**C**) amount of sorbent, (**D**) extraction time, (**E**) salt addition, and (**F**) sample pH.

**Figure 3 molecules-24-02758-f003:**
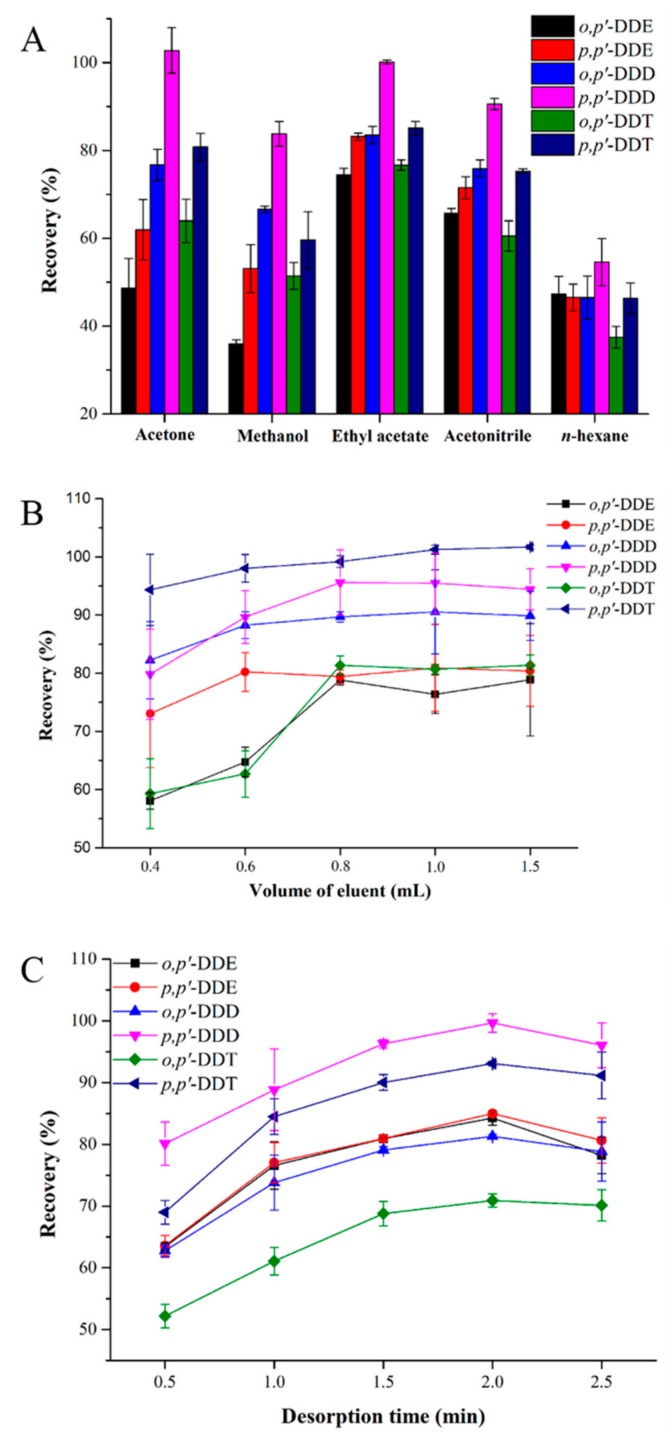
Effect of different desorption conditions on the desorption performance of DDTs: (**A**) type of eluent, (**B**) volume of eluent, and (**C**) desorption time.

**Figure 4 molecules-24-02758-f004:**
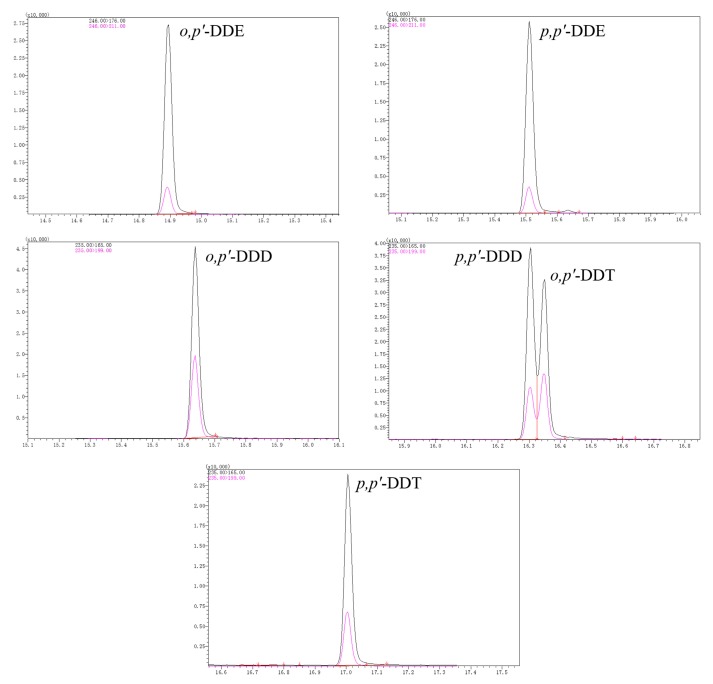
Extracted total ion chromatograms obtained by the MM/ZIF-8/IL-based method for DDTs from river water spiked at 10 µg L^−1^.

**Figure 5 molecules-24-02758-f005:**
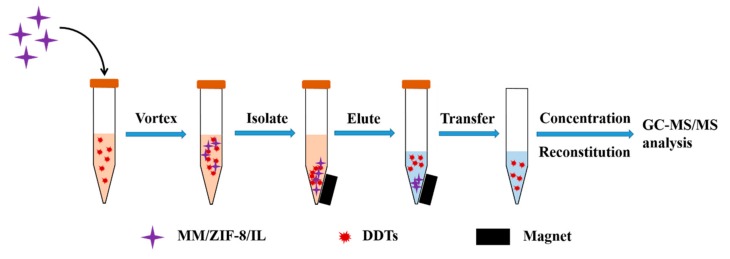
Typical procedure of the proposed magnetic solid-phase extraction (MSPE) method.

**Table 1 molecules-24-02758-t001:** Analytical parameters of the MM/ZIF-8/IL-MSPE method for the analysis of DDTs from ultrapure water samples.

DDTs	Calibration Equation	Linear Range (µg L^−1^)	*R* ^2^	LOD (µg L^−1^)	RSD^a^ (%) (*n* = 6)	
					Intraday	Interday
*o,p′*-DDE	y = 4916.2x − 39626	0.5–500	0.9956	0.0016	5.5	7.1
*p,p′*-DDE	y = 4303.6x − 28468	0.5–500	0.9971	0.0021	2.3	4.5
*o,p′*-DDD	y = 7825.8x − 64397	0.5–500	0.9955	0.0034	1.0	1.0
*p,p′*-DDD	y = 7345.3x − 73748	0.5–500	0.9934	0.0033	6.5	8.9
*o,p′*-DDT	y = 6887.9x − 54005	0.5–500	0.9953	0.0032	2.9	5.4
*p,p′*-DDT	y = 5730.4x − 59956	0.5–500	0.9927	0.0072	1.9	6.3

LOD: limit of detection; RSD: relative standard deviation

**Table 2 molecules-24-02758-t002:** Comparison of different methods for analysis of DDTs in water samples.

Method	Sorbent	Sample Amount (mL)	Sorbent Amount (mg)	Extraction Time (min)			Type and Volume of Eluent (mL)	Desorption Time (min)	Linear Range	LOD	Ref.
MSPE-GC-MS/MS	Magnetoliposome	400	140	20			Acetonitrile, 3.0; acetone, 3.0	2	1–125 ng L^−1^	0.35 ng L^−1^	[40]
μ-SPE-GC-MS	MIL-101	10	-	40			Ethyl acetate, 0.1	15	0.05–50 μg L^−1^	0.003 µg L^−1^	[41]
MSPE-GC-MS	BMZIF-derived porous carbon ^a^	10	6	10			Dichloromethane, 2	10	2–500 ng L^−1^	0.39–0.65 ng L^−1^	[9]
MSPE-GC-MS/MS	M-M-ZIF-67 ^b^	5	6	20			Acetonitrile, 4	10	1–200 µg L^−1^	0.07–1.03 µg L^−1^	[17]
MSPE-GC-MS/MS	MM/ZIF-8/IL	10	4	1.5			Ethyl acetate, 1.6	4	0.5–500 µg L^−1^	0.0016–0.0072 µg L^−1^	This work

^a^ Magnetic porous carbon derived from bimetallic metal–organic framework. ^b^ Magnetic multiwalled carbon nanotubes/zeolitic imidazolate framework-67.

**Table 3 molecules-24-02758-t003:** Analytical results for determination of DDTs in real water samples.

Matrix	Analyte	Spiked Concentration (µg L^−1^) (*n* = 3)				
		0	10		100	
		Found	Recovery (%)	RSD (%)	Recovery (%)	RSD (%)
Tap water	*o,p′*-DDE	ND. ^a^	87.8	2.7	93.7	1.5
	*p,p′*-DDE	ND.	95.3	2.7	95.4	2.3
	*o,p′*-DDD	ND.	95.0	2.5	94.3	2.2
	*p,p′*-DDD	ND.	90.0	2.9	93.4	2.2
	*o,p′*-DDT	ND.	81.0	4.1	85.5	3.4
	*p,p′*-DDT	ND.	93.1	0.6	97.5	0.5
River water	*o,p′*-DDE	ND.	79.8	1.4	88.2	2.6
	*p,p′*-DDE	ND.	89.4	1.8	93.9	2.1
	*o,p′*-DDD	ND.	81.6	1.2	85.1	1.4
	*p,p′*-DDD	ND.	96.6	1.8	98.5	0.7
	*o,p′*-DDT	ND.	82.2	1.7	90.7	1.5
	*p,p′*-DDT	ND.	84.7	1.7	91.2	1.6
Underground water	*o,p′*-DDE	ND.	87.3	0.4	89.6	1.1
	*p,p′*-DDE	ND.	85.4	1.7	95.4	0.8
	*o,p′*-DDD	ND.	95.1	1.9	96.8	1.0
	*p,p′*-DDD	ND.	84.7	3.3	85.7	1.6
	*o,p′*-DDT	ND.	72.6	1.9	84.7	1.8
	*p,p′*-DDT	ND.	87.7	1.8	91.8	0.8

^a^ ND means not detected.

**Table 4 molecules-24-02758-t004:** Acquisition and chromatographic parameters for DDTs.

DDTs	*t*_R_ (min)	MRM1 (*m*/*z*)	CE1 ^a^ (eV)	MRM2 (*m*/*z*)	CE2 (eV)
*o,p′*-DDE	14.943	246.00 > 176.00	30	246.00 > 211.00	22
*p,p′*-DDE	15.560	246.00 > 176.00	30	246.00 > 211.00	22
*o,p′*-DDD	15.600	235.00 > 165.00	24	235.00 > 199.00	14
*p,p′*-DDD	16.350	235.00 > 165.00	24	235.00 > 199.00	14
*o,p′*-DDT	16.402	235.00 > 165.00	24	235.00 > 199.00	16
*p,p′*-DDT	17.058	235.00 > 165.00	24	235.00 > 199.00	16

^a^ CE: collision energy.

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
