# Peer review of "Magnetic Solid-Phase Extraction of Dichlorodiphenyltrichloroethane and Its Metabolites from Environmental Water Samples Using Ionic Liquid Modified Magnetic Multiwalled Carbon Nanotube/Zeolitic Imidazolate Framework-8 as Sorbent"

_molecules, 2019, doi:10.3390/molecules24152758_

Reviewer 1 Report

The Experimental work is well carried out. The conclusions are appropriate and in my opinion the work can be published as is.

Author Response

Thank you very much for this review. In view of your comments for this manuscript,  we hold the opinion that no more response for revision is needed in here. 

Reviewer 2 Report

In this work, a four-component composite composed of iron oxide, MWCNT, ZIF-8, and an ionic liquid is used as a host material for adsorbing various DDT derivatives in water. The composite adsorbs DDTs quickly, and the adsorbed DDTs are desorbed in neat organic solvent, then are subject to microanalysis. The results indicate that this analytical method called magnetic solid-phase extraction (MSPE) is simple and effective for detecting very small amounts of harmful organic molecules in water.

Although the role of each component, especially ZIF-8 and ionic liquid is vague, it seems that the technique presented in this work may be available for analytical and environmental chemistry. So, I would like to support the publication of this work.

Other comment.

1. ZIF-8 has small pore opening, and therefore, any DDT derivatives may not access the internal pore of ZIF-8 because the molecular sizes of the DDTs are much greater than the pore openings. Therefore, the authors should examine in more detail the role of ZIF-8 when the authors really want to develop the composite for practical applications.  

Author Response

Thank you very much for this good suggestion. Honestly, we totally agree with this constructive comment. For the developing of the composite for practical applications in future, ZIF-8 needs to be placed in a kernal role. Meanwhile, adsorption mechanism of ZIF-8 composite for target analytes needs to be investigated either.